Corrected: Author correction

# TriQuinoline

Shinya Adachi [1], Masakatsu Shibasaki[1] & Naoya Kumagai [1]

The bottom-up synthesis of structurally well-defined motifs of graphitic materials is crucial to understanding their physicochemical properties and to elicit new functions. Herein, we report the design and synthesis of TriQuinoline (TQ) as a molecular model for pyridinic-nitrogen defects in graphene sheets. TQ is a trimer of quinoline units concatenated at the 2- and 8-positions in a head-to-tail fashion, whose structure leads to unusual aromatisation behaviour at the final stage of the synthesis. The central atomic-sized void endows TQ with high proton affinity, which was confirmed empirically and computationally. TQ•H$^+$ is a two-dimensional cationic molecule that displays both π–π and CH–π contact modes, culminating in the formation of the ternary complex ([12]cycloparaphenylene(CPP) ⊃ (TQ•H$^+$/coronene)) that consists of TQ•H$^+$, coronene (flat), and [12]cycloparaphenylene ([12]CPP) (ring). The water-miscibility of TQ•H$^+$ allows it to serve as an efficient DNA intercalator for e.g. the inhibition of topoisomerase I activity.

[1] Institute of Microbial Chemistry, 3-14-23 Kamiosaki, Shinagawa-ku, Tokyo 141-0021, Japan. Correspondence and requests for materials should be addressed to N.K. (email: nkumagai@bikaken.or.jp)

Molecular aesthetics is the primary driving force for the intuitive design of xenobiotic molecular entities, which has captivated chemists in the pursuit of novel chemical structures to elicit hitherto unknown physicochemical properties[1]. The rapidly growing applications of graphitic materials and polyaromatic hydrocarbons (PAHs) are spurring research on exotic polyaromatic molecules that feature skewed and non-flat three-dimensional (3D) architectures (Fig. 1a)[2–13]. Another area of intense research is the modification of the physicochemical properties of PAHs by strategic doping with heteroatoms[14–17]. Recent advances in analytical techniques have enabled the detailed analysis of individual heteroatom dopants in graphitic materials, which allows for identification of the distinctive nature of graphitic- and pyridinic-nitrogen sites[18–21]. Various methods have emerged for the preparation of graphitic materials with structural defects surrounded by three pyridinic nitrogens[22–24], paving the way for the introduction of metal cations to impart, e.g., particular catalytic functionality[25,26] with prospective applications in energy conversion[27] and biocompatible catalysis[28]. However, the introduction of structurally defined defects is often random, and precise control over the position of such defects remains elusive. We were therefore particularly interested in a bottom-up and structurally defined synthesis of miniaturised graphitic-nitrogen sites that feature an inherent atomic-sized void that may potentially display the characteristic physicochemical properties of small molecules amenable to a variety of spectroscopic analysis techniques.

In the pursuit of such a model compound with minimal two-dimensional (2D) size, herein we describe TriQuinoline (TQ), a molecule that comprises three quinoline units concatenated at the 2- and 8-positions in a head-to-tail fashion (Fig. 1b). TQ represents a pseudo-$C_{3h}$-symmetric virtually flat molecule with a central defect surrounded by three chemically identical $sp^2$-hybridised nitrogen atoms. The applied indirect synthesis revealed an unusually rapid intermolecular hydride transfer that favours the formation of a quinoline via facilitated aromatisation. The atomic-size void located at the centre of the rigid 2D TQ scaffold displays remarkably high proton affinity, which in turn endows the molecule with water miscibility and the ability to engage in directional complexation with other graphitic materials via π (cation)–π and CH–π interactions.

## Results

**Synthesis of TQ**. Initially, we envisioned a retrosynthetic path that involves the disconnection of the three C–C bonds that interconnect the three quinoline units, which could then be linked via transition-metal-catalysed cross-coupling reactions of plausible intermediates that bear suitable functional groups (Fig. 1c). Indeed, acyclic intermediates that contain three quinoline units were readily accessible by standard synthetic protocols. However, the formation of the last C–C bond was unsuccessful, presumably owing to the rigid flat architecture of TQ, which is probably mismatched with the transition-state structure required for the cross-coupling to proceed. Hence we explored a mechanistically distinct reaction, employing a lithiated species to promote nucleophilic cyclisation, followed by spontaneous oxidative aromatisation to afford TQ (Supplementary Methods). Contrary to the expected lipophilic nature ($C\log P =$ 5.96), TQ was isolated solely in the water layer after aqueous work-up, which implies that TQ is readily protonated under neutral conditions. Although the identity of TQ was confirmed after reverse-phase high-performance liquid chromatography purification and lyophilisation, the extremely poor yield (5.7%) prompted us to devise an alternative synthetic strategy. We envisioned that intramolecular imine formation would be feasible

given the absence of structural constraints and that a subsequent wall-reinforcement with an ethylene fragment could furnish the TQ framework (Fig. 1c).

On the basis of this assumption, we devised a corresponding synthetic protocol (Fig. 2). The Suzuki–Miyaura cross-coupling[29] of 2,8-dichloroquinoline **1** and N-Boc-functionalised phenylboronic acid **2**, in which the regioselective introduction of an aniline unit was achieved, afforded **3**. Subsequently, the residual C–Cl bond was replaced with a C–B bond using Molander's borylation protocol[30,31], before the thus-obtained intermediate **4** was directly subjected to a second Suzuki–Miyaura cross-coupling with 2-chloro-8-methylquinoline **5** to furnish **6** in 66% yield over two steps. An ensuing two-fold benzylic radical bromination generated dibromide **7**, which was exposed to aqueous $Na_2CO_3$ at 80 °C to deliver aldehyde **8**. Removal of the N-Boc group with trifluoroacetic acid (TFA) led to the formation of the TFA salt of cyclic diquinoline imine (DQ-Im•TFA) **9**. From **6** to DQ-Im•TFA **9**, each step proceeded smoothly in almost quantitative conversion, which allowed the products to be utilised directly in consecutive steps without purification. n-Butyl vinyl ether (BVE) afforded the two requisite C–C bonds by gentle heating with DQ-Im•TFA **9** via a formal inverse-electron-demand hetero-Diels–Alder (IED-HDA) reaction, followed by a spontaneous elimination/aromatisation sequence, which furnished the desired TQ architecture in 46% yield in four steps. TQ•TFA **10** was extracted into the aqueous phase, and a subsequent lyophilisation allowed its isolation in pure form. The $^1$H NMR spectrum matched that of the sample obtained via the lithiated intermediate (Supplementary Fig. 1). The resonance of the central proton was observed at >22 ppm, which is comparable to the chemical shift of structurally related macrotricyclic aminopyridines[32].

**Mechanistic details of the unusual quinoline formation**. Notably, the last step of the synthesis of TQ•TFA **10** exhibits unexpected behaviour compared to typical Povarov reactions. The Povarov reaction[33,34], which was reported over 50 years ago, is classified as a formal IED-HDA reaction, specifically in the case of N-aryl imines to construct a tetrahydroquinoline (THQ) framework. To our surprise, any traces of the plausible primary and secondary products after elimination, i.e., THQ and 1,2-dihydroquinoline (DHQ), were not detected in the reaction of DQ-Im•TFA **9** and BVE (Supplementary Methods). $^1$H NMR monitoring of the reaction performed under rigorously inert conditions (degassed $CD_3CN$ at 25 °C under Ar in a dry box) allowed for detection of the complete consumption of DQ-Im•TFA **9** after 25 h with concomitant formation of a ca. 1:1 mixture of TQ•TFA **10** and the reduced DiQuinoline-Amine•TFA (DQ-Am•TFA) **11** (Fig. 3a, Supplementary Fig. 9). This observation confirmed the rapid elimination of n-butanol and that DQ-Im•TFA **9** serves not only as a substrate, but also as a hydride acceptor, thus transforming the intermediate DHQ into TQ•TFA. Indeed, DQ-Im•TFA **9** was readily reduced by a Hantzsch ester (25 °C, 10 min), i.e., a typical organic reductant, to generate DQ-Am•TFA **11** (Supplementary Fig. 7), which demonstrates the ability of DQ-Im•TFA **9** to accept a hydride from DHQ to furnish the TQ architecture (Supplementary Methods). Exposure of the aforementioned 1:1 mixture to an ambient atmosphere led to the gradual emergence of complex signals in the $^1$H NMR spectrum which converged into a single set of signals attributed to TQ•TFA **10**, indicating that DQ-Am **11** is susceptible to oxidation by air, resulting in the reversion to DQ-Im•TFA **9** (Supplementary Figs. 7 and 8). This notion was confirmed by monitoring the reaction at 25 °C in air by $^1$H NMR spectroscopy, which revealed divergent concentrations of DQ-Im•TFA **9** and DQ-Am•TFA **11** under steady formation of

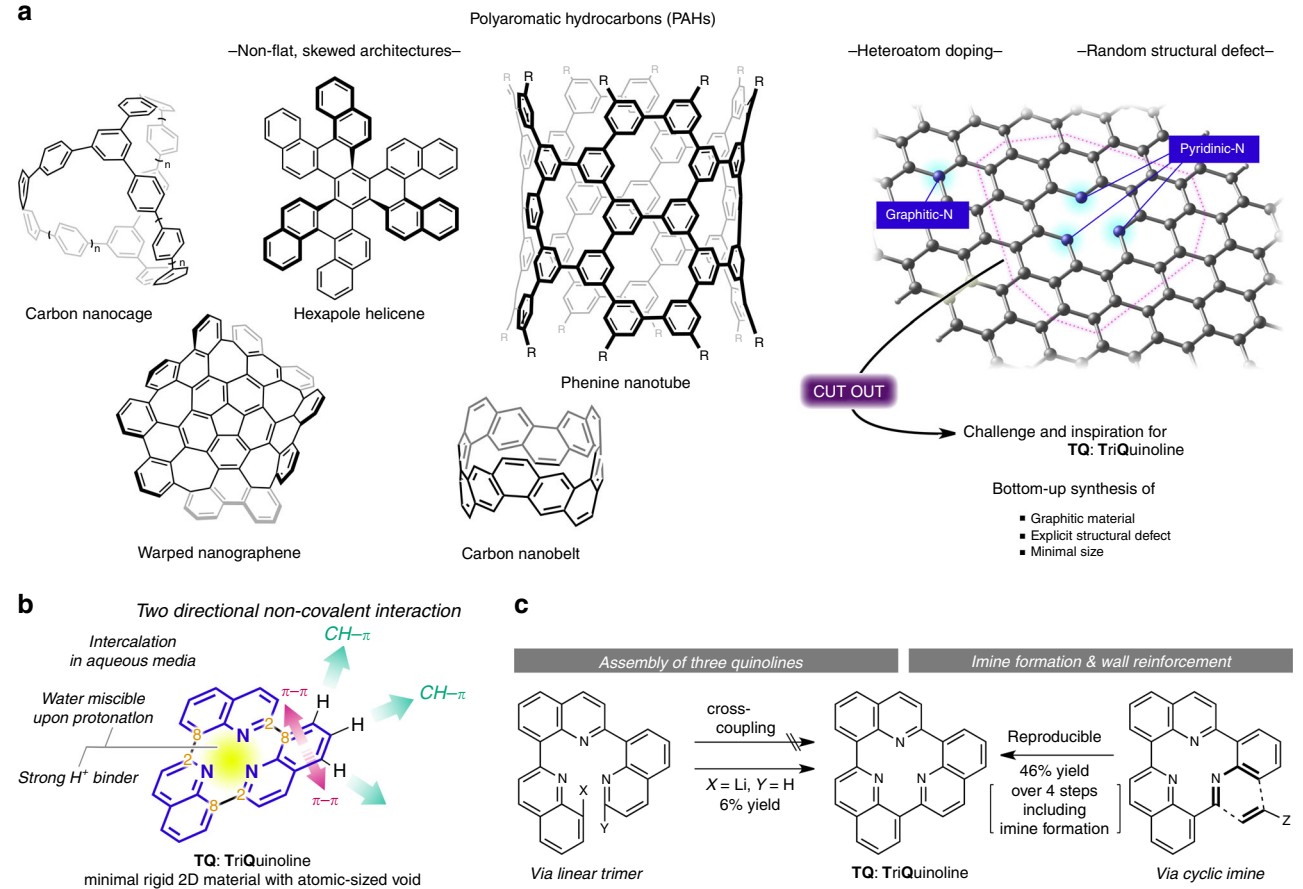

**Fig. 1** Overview and inspiration for this work. **a** State-of-the-art on polyaromatic hydrocarbon research. Structures of seminal non-flat and skewed graphitic materials: carbon nanobelts and hexapole helicene. Arbitrary doping of nitrogen atoms in PAHs furnishes graphitic nitrogen and pyridinic-nitrogen centres, whereby the latter affords a one-atom defect. Bottom-up synthesis of a miniaturised graphitic material with pyridinic nitrogens corresponding to TQ: TriQuinoline. **b** Overview of the structural and chemical properties of TQ. **c** Unsuccessful attempts to synthesise TQ and an alternative viable synthetic approach

**Fig. 2** Synthesis of TQ: TriQuinoline. Purification was performed at stages **3** and **6**. The final step represents the unusual Povarov reaction, in which plausible tetrahydro- and dihydroquinoline intermediates were undetectable, furnishing the TQ architecture without the need for additional elimination and oxidation conditions. AIBN azobisisobutyronitrile, NBS *N*-bromosuccinimide, TFA trifluoracetic acid

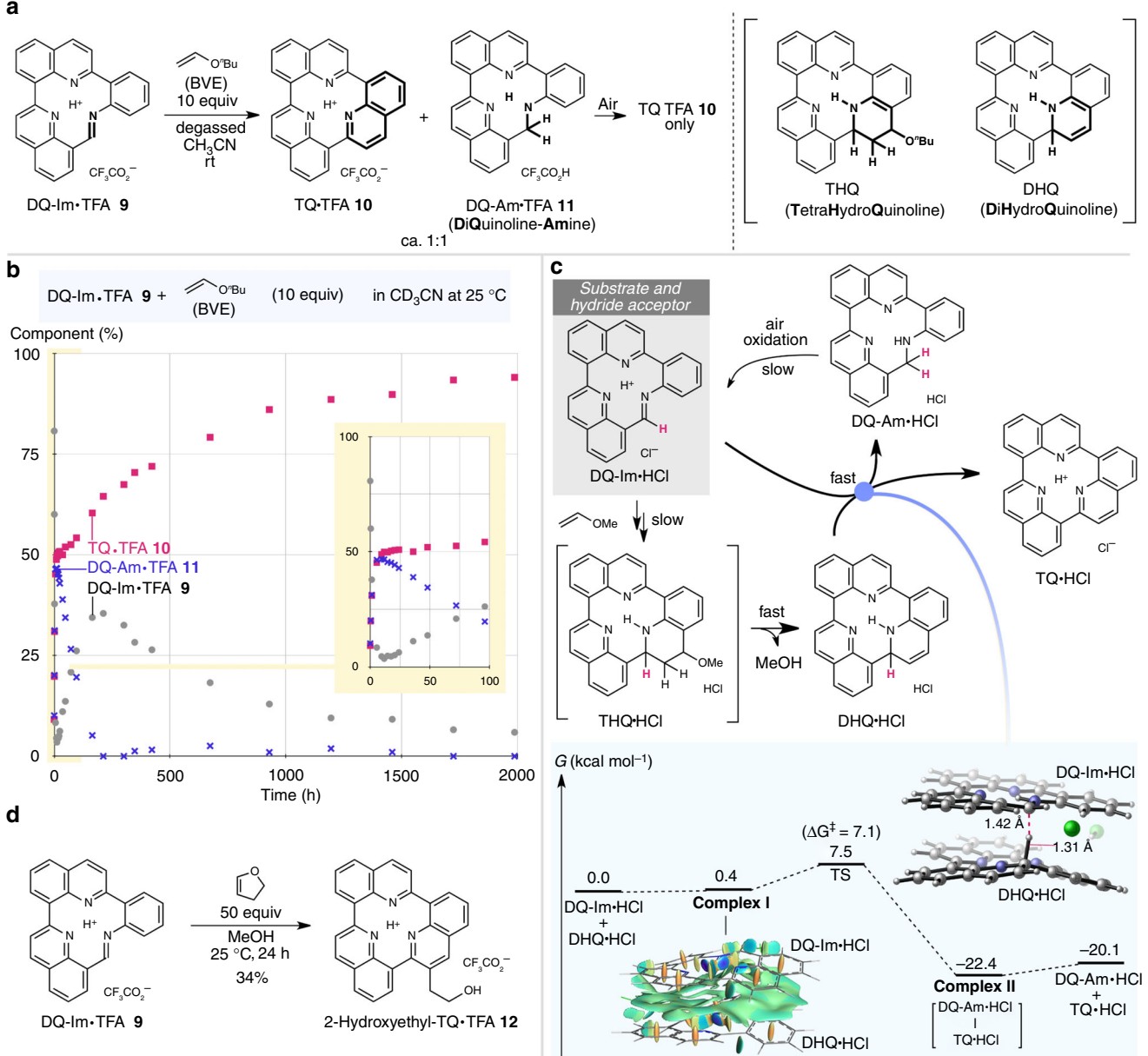

**Fig. 3** An unusual non-stop Povarov reaction allows the one-pot construction of the quinoline unit en route to TQ. **a** Evaluation of the unusual Povarov reaction of DQ-Im•TFA **9** and BVE under degassed conditions, affording a ca. 1:1 mixture of TQ•TFA **10** and DQ-Am•TFA **11**. Tetrahydroquinoline (THQ) or 1,2-dihydroquinoline (DHQ) intermediates were not detected. **b** Reaction profile of DQ-Im•TFA (1 equiv) and BVE (10 equiv) in CD₃CN monitored at room temperature by ¹H NMR analysis. Inset: early stage of the reaction (0–100 h). **c** Plausible reaction mechanism based on the model reaction of DQ-Im•HCl and methyl vinyl ether calculated at the ωB97XD/def2-TZVPP/SMD(acetonitrile)//B3LYP-D3/6-31G(d) level of theory. The energy diagram of the hydride transfer step is shown. The full energy landscape of the sequential reactions is provided in Supplementary Fig. 50. Non-covalent interactions visualised by the NCIPLOT programme. The blue and green areas denote strong and weak attractive interactions, respectively. **d** Synthesis of TQ derivative **12** with a 2-hydroxyethyl tail. BVE *n*-butyl vinyl ether

TQ•TFA **10** (Fig. 3b, Supplementary Figs. 5 and 6). While BVE underwent undesired competitive hydrolysis to acetaldehyde at 25 °C and was consumed almost entirely within 25 h (Supplementary Fig. 5), the regenerated DQ-Im•TFA **9** slowly reacted even with the significantly less reactive acetaldehyde, resulting in a constant generation of TQ•TFA over time (Fig. 3b, Supplementary Figs. 12 and 13). To further investigate the unusual behaviour of such a continuous Povarov reaction, we carried out density functional theory (DFT) calculations at the ωB97XD/def2-TZVPP/SMD(acetonitrile)//B3LYP-D3/6-31G(d) level of

theory in order to define the energy profile using DQ-Im•HCl and methyl vinyl ether (MVE) as model substrates (Supplementary Figs. 50 and 51)[35–38]. The initial 1,2-addition of MVE to DQ-Im•HCl, followed by Friedel–Crafts cyclisation to give the THQ intermediate was identified as the most energy-demanding step ($\Delta G^{\ddagger} = 20.5$, 20.2 kcal mol⁻¹). The energy barriers of the subsequent steps, i.e., the elimination of MeOH to afford DHQ ($\Delta G^{\ddagger} = 14.7$ kcal mol⁻¹) followed by hydride transfer from DHQ to DQ-Im ($\Delta G^{\ddagger} = 7.1$ kcal mol⁻¹), were estimated as sufficiently lower than that of the addition of MVE, rendering the THQ and

DHQ intermediates virtually undetectable (Fig. 3c, Supplementary Fig. 5). The remarkably facilitated hydride transfer likely originates from the favourable formation of plane-to-plane complex **I**, which comprises DHQ and DQ-Im via attractive π-contacts, as qualitatively visualised by NCIPLOT ($\Delta E = -30.2$ kcal mol$^{-1}$ after basis set superposition error (BSSE) correction; Supplementary Fig. 52)[39]. Inspection of the intrinsic reaction coordinate path revealed a change in the plane-to-plane distance of merely 2.0% of the travelling distance of the hydride (Supplementary Fig. 54), affording complex **II**, which comprises TQ and reduced amine DQ-Am. This unusual non-stop Povarov-type reaction was ascribed to the unique 2D structure of DQ-Im•TFA **9** and found to be also valid for the reaction using a different enol ether, e.g. 1,2-dihydrofuran, giving TQ derivative **12** (Supplementary Methods), which bears a hydroxyethyl group on the periphery that is amenable to further functionalisation (Fig. 3d).

**Physicochemical properties of TQ.** With a reliable and reproducible synthetic protocol for TQ•TFA **10** in hand, we then turned our attention to the physicochemical properties of the TQ architecture. Numerous attempts at removing the proton captured at the centre of the molecule were unsuccessful, and partial decomposition under basic conditions afforded an unidentified mixture of compounds (e.g. aqueous phase: NaOH, in acetonitrile: from weak Et$_3$N to strong *tert*-butylimino-tri(pyrrolidino)phosphorene). Intuitively, the central void of the proton-free TQ is expected to be unstable due to repulsive overlap of the lone-pair orbitals of the three nitrogen atoms. Given that the three quinoline units are interconnected to form TQ with no topological constraints (bond angles and lengths), a computational evaluation of hypothetical homodesmotic reactions[40] allowed the quantification of the negative contribution of the central part to the thermodynamic stability of TQ (Fig. 4a, Supplementary Methods, Supplementary Figs. 55 and 56). Proton-free TQ was found to be much less stable than the open form ($\Delta H = -25.1$ kcal mol$^{-1}$), presumably due to repulsive interactions between the three nitrogen atoms, which is corroborated by the markedly different calculated energy for the corresponding homodesmotic reaction of protonated TQ•H$^+$ ($\Delta H = 6.4$ kcal mol$^{-1}$). Indeed, the calculated absolute proton affinity of TQ (277.0 kcal mol$^{-1}$) is significantly higher than that of reference compound 1,8-bis(dimethylamino)naphthalene superbase (proton sponge) **13** (245.3 kcal mol$^{-1}$) at the same level of theory (Supplementary Methods)[41]. The kinetic reluctance of TQ•H$^+$ towards proton exchange was experimentally probed by $^1$H NMR spectroscopy (Fig. 4b). In sharp contrast to the rapid decay of the proton signal of **13**•TFA in CD$_3$OD (<30 min), the resonance of TQ•TFA **10** persisted for a significantly longer period (>10% of H$^+$ remained even after 1 week) (Fig. 4b, Supplementary Methods, Supplementary Figs. 14–29). Although TQ•H$^+$ and DQ-Im•H$^+$ share structural characteristics, specifically the size of the cavity surrounded by three $sp^2$-hybridised nitrogen atoms, the retention time of the captured proton was drastically different, i.e., DQ-Im•H$^+$ released the proton virtually instantaneously. Molecular dynamics simulations on TQ•H$^+$ and DQ-Im•H$^+$ by the atom-centred density matrix propagation (ADMP) method suggested that TQ•H$^+$ is structurally more rigid than DQ-Im•H$^+$, which renders the proton captured by the former kinetically less labile (Supplementary Methods, Supplementary Fig. 57)[42].

The discernible π-contacts found in complex **I** (DHQ and DQ-Im in Fig. 3c) by DFT calculations at the hydride transfer step suggest that TQ•H$^+$ is capable of emulating non-covalent interactions to build supramolecular entities (Supplementary Methods)[43]. Coronene, a neutral π-sheet with a 2D size similar to that of TQ, exhibited a characteristic upfield shift (9.10 to 8.82

ppm) of its $^1$H NMR (DMSO-$d_6$) spectrum in the presence of TQ•TFA **10**, corroborating the formation of plane-to-plane π-complex TQ/coronene **14** (Fig. 5a, b, e)[44]. Intriguingly, TQ•TFA **10** was also able to form a binary complex via edge-to-plane contacts with [12]cycloparaphenylene ([12]CPP), a ring-shaped π-wall molecule (Fig. 5a, c, f)[45–47]. In contrast to the case of coronene, the signal derived from [12]CPP experienced a downfield shift (from 7.76 to 7.90 ppm), indicating that TQ•H$^+$ is able to magnetically deshield the proton resonance of [12]CPP, similar to the case of [10]CPP ⊃ C$_{60}$ inclusion complexes[48,49]. As expected, the complexation of TQ and CPP is size- and shape-specific; no evidence of complexation by $^1$H NMR analysis was found for the combination of the smaller [10]CPP ring and TQ•TFA **10** (Supplementary Fig. 44), or that of [12]CPP and 2-hydroxyethyl-TQ•TFA **12** (Fig. 5b, i, j, k, Supplementary Fig. 43). These observations strongly imply that the complex presents a spoke-and-rim topology and that CH–π interactions are the primary source of attraction[50–52]. Recurrently, stark differences were observed between TQ and DQ-Im; DQ-Im•TFA **9** failed to form a similar inclusion complex with [12]CPP, even though TQ and DQ-Im share a similar 2D structure (Fig. 5g, h, i, Supplementary Fig. 41). NMR titration together with non-linear regression analysis revealed the formation of the 1:1 and 1:2 inclusion complexes [12]CPP ⊃ TQ•H$^+$ **15** and [12]CPP ⊃ (TQ•H$^+$)$_2$ **16**, respectively, both of which exhibit comparable association constants $K_1 = (1.20 \pm 0.035) \times 10^3$ M$^{-1}$ and $K_2 = (1.07 \pm 0.011) \times 10^3$ M$^{-1}$ (Fig. 5c, Supplementary Fig. 34)[53–55]. An electrospray ionisation time-of-flight (ESI-TOF) MS analysis provided further experimental support for the formation of inclusion complexes **15** and **16** as well as plane-to-plane complex **14** (Supplementary Figs. 30 and 32).

The favourable interactions in these π-rich molecules were competently reproduced by simulations using density functionals adopting empirical dispersion corrections (Supplementary Methods). Calculations on the cationic complex [12]CPP ⊃ TQ•H$^+$ **15** without a counter anion at the ωB97XD/def2-TZVPP/SMD (acetonitrile)//B3LYP-D3/6-31G(d,p) level of theory provided the free energy of association ($\Delta G^{\ddagger} = -7.94$ kcal mol$^{-1}$), while the attractive interactions appeared to be overestimated (Supplementary Fig. 58). Decomposition of the interaction energy by symmetry-adapted perturbation theory revealed that dispersion interactions are the main contributor towards the formation of the inclusion complex (Supplementary Fig. 66)[56]. The upfield and downfield shifts in the spectra of TQ•H$^+$ and [12]CPP, respectively, were qualitatively reproduced through estimation of the shielding effect by gauge-independent atomic orbital calculations (Supplementary Fig. 65). The uniform upfield shift of the proton resonance of TQ•H$^+$ is thereby consistent with the previous literature on the structurally related CH–π-driven inclusion complex. Isobe et al.[57,58] reported an inclusion complex featuring a bowl-in-tube topology comprising corannulene (bowl) and cyclic chrysene tetramer (tube), where 10 CH–π hydrogen bonds are responsible for the formation of the inclusion complex. To further probe the potential contribution of CH–π interactions to the formation of [12]CPP ⊃ TQ•H$^+$ **15**, we performed a topological analysis on the electron density ($\rho(r_c)$) of the complex in the context of Bader's quantum theory of atoms-in-molecules (QTAIM)[59–61]. A total of 15 intermolecular bond paths between the peripheral hydrogens of TQ•H$^+$ and proximal carbons of [12]CPP were found together with [3, –1] bond critical points (BCPs) (Fig. 6a). The electron density $\rho(r_c)$, the Laplacian of the electron density ($\nabla^2\rho(r_c)$), and the ratio of the curvature of the density at the BCPs ($|\lambda_1|/\lambda_3$) have been proposed as the principal descriptors to characterise CH–π interactions, and typically fall in the ranges of $0.002 < \rho(r_c) < 0.0034$, $0.02 < \nabla^2\rho(r_c) < 0.14$, and $|\lambda_1|/\lambda_3 < 1$, respectively[57,62,63]. Among the 15 intermolecular BCPs found,

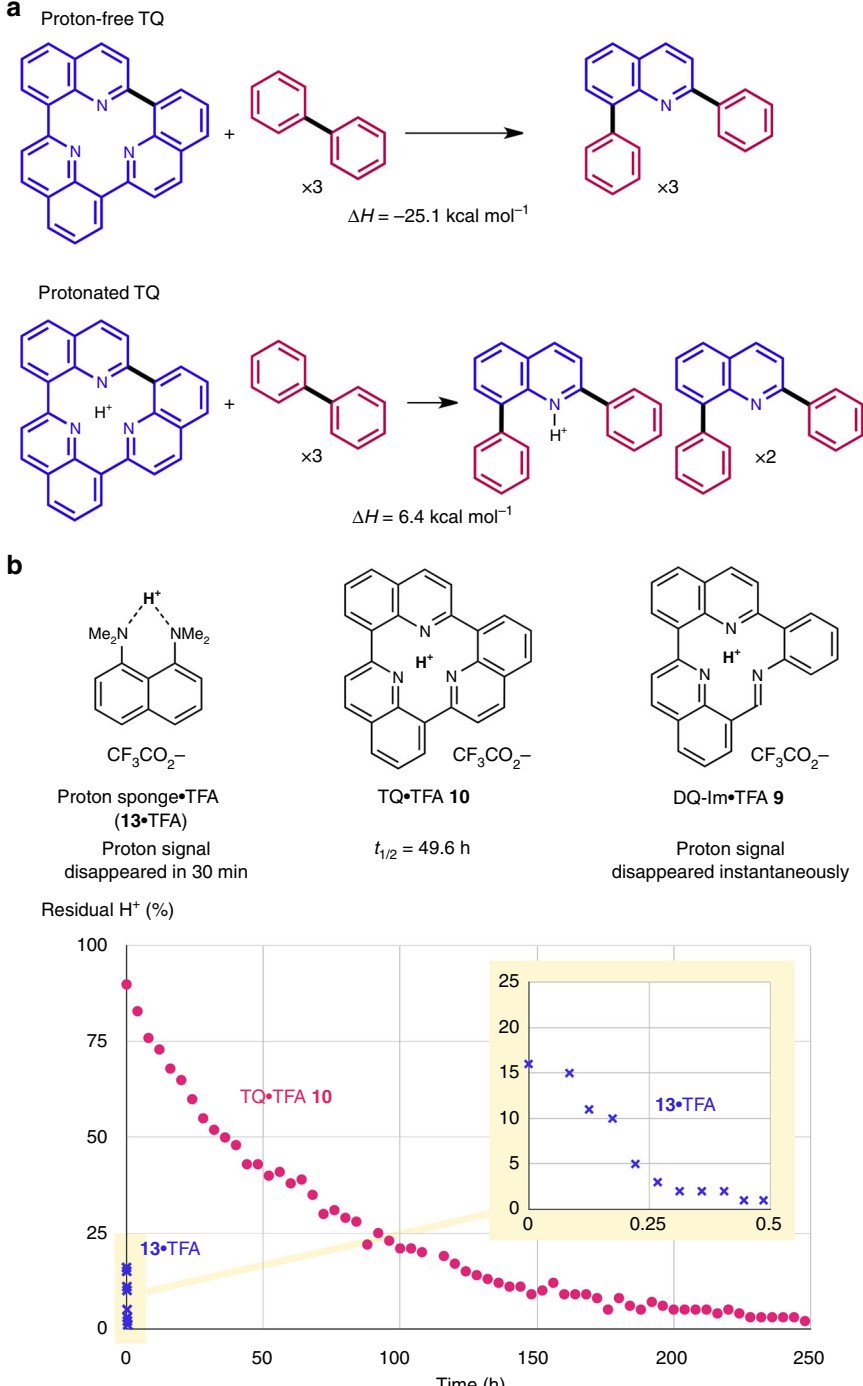

**Fig. 4 Unparalleled protophilicity of TQ. a** Homodesmotic reactions of proton-free TQ and protonated TQ. Heat-of-formation values were calculated at the ωB97XD/def2-TZVPP/SMD(acetonitrile)//B3LYP-D3/6-31 G(d, p) level of theory. **b** Retention of the proton of **13•**TFA, TQ•TFA **10**, and DQ-Im•TFA **9** in CD₃OD (>10⁴ equiv with respect to the solute) at 25 °C. Kinetic profile of the H⁺/D⁺ exchange monitored for **13•**TFA and TQ•TFA **10** by ¹H NMR analysis. Inset: Magnification of the early stage of the reaction (0–0.5 h) for **13•**TFA

9 of these involve the protons at the 4-, 5-, and 6-positions of each quinoline unit that are geometrically close to the π-plane of [12] CPP and satisfy the aforementioned criteria, which strongly suggests that CH–π interactions play a dominant role in the formation of inclusion complex **15** (Fig. 6a). The presence of CH–π interactions was also determined for [12]CPP ⊃ (TQ•H⁺)₂ **16** by QTAIM analysis, where the BCPs interconnecting protons at the 5- and 6-positions of each quinoline meet the criteria, and the plane-to-plane distance (3.35 Å) between two TQ•H⁺ molecules

is indicative of attractive π-contacts. The observation of plane-to-plane and edge-to-plane interactions prompted us to construct a ternary inclusion complex composed of three different molecules. The incremental addition of coronene to a mixture of [12]CPP and TQ•TFA in DMSO-d₆ led to a significant enhancement of the upfield and downfield shifts of the coronene and [12]CPP signals, respectively, while the resonance of TQ•TFA were shifted even further upfield (Fig. 5d). Assuming the absence of complexation between [12]CPP and coronene due to a size mismatch

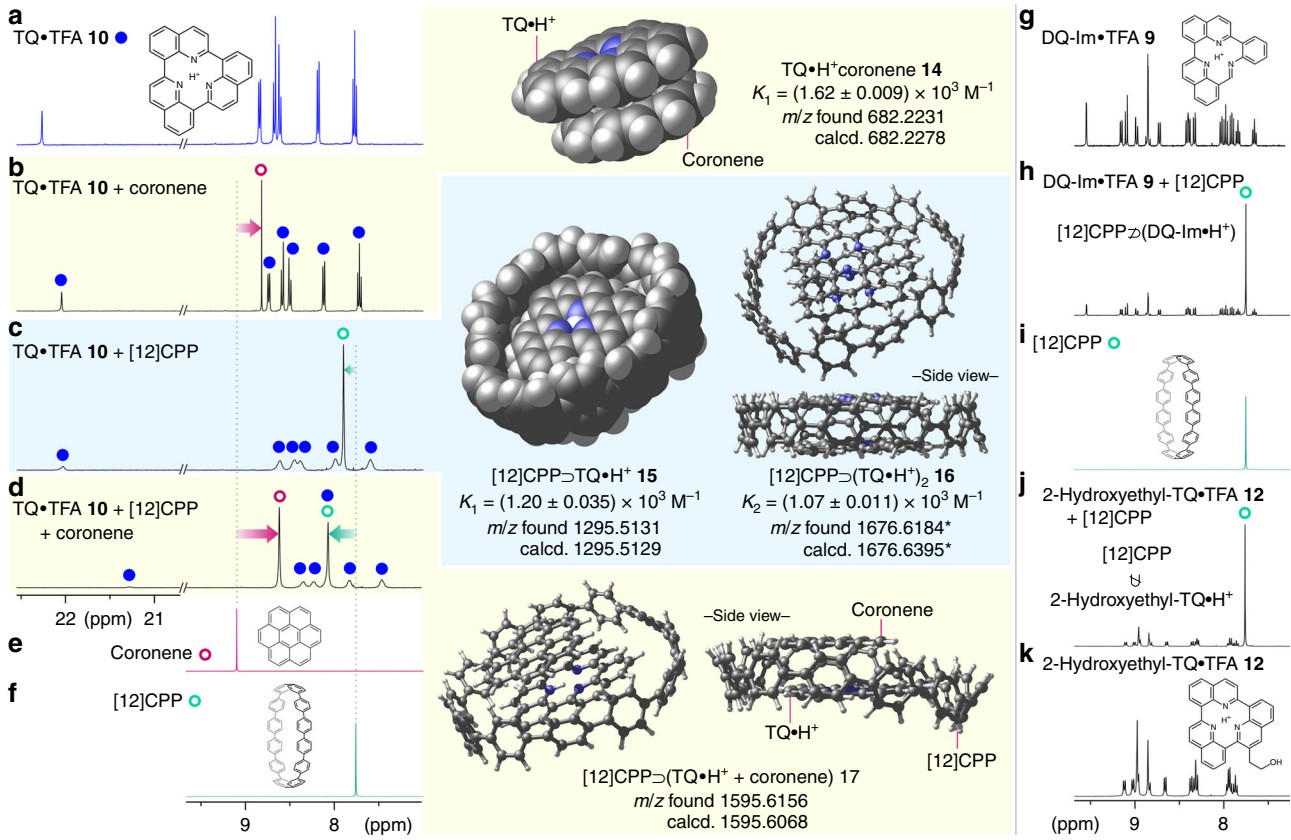

**Fig. 5** Complexation properties of TQ via plane-to-plane and edge-to-plane non-covalent π-interactions. **a–f** Expanded $^1$H NMR spectra (DMSO-$d_6$) of TQ•TFA **10** (**a**), TQ•TFA **10** and coronene (5:1) (**b**), TQ•TFA **10** and [12]CPP (5:1) (**c**), TQ•TFA **10**, [12]CPP, and coronene (5:1:5) (**d**), coronene (**e**), and [12]CPP (**f**). For **b–d**, the optimised structures of the corresponding complexes, calculated at the B3LYP-D3/6-31G(d,p) level of theory, are shown as either space-filling (vDW) or ball-and-stick models together with supportive data from the ESI-TOF MS analysis. For **b** and **c**, the association constants were determined by NMR titration. **g–k** Expanded $^1$H NMR spectra (DMSO-$d_6$) of DQ-Im•TFA **9** (**g**), DQ-Im•TFA **9** and [12]CPP (5:1) (**h**), [12]CPP (**i**), 2-hydroxyethyl-TQ•TFA **12** and [12]CPP (5:1) (**j**), and 2-hydroxyethyl-TQ•TFA **12** (**k**). No discernible spectral changes were observed in the case of DQ-Im•TFA, 2-hydroxyethyl-TQ•TFA **12**, and [12]CPP, indicating the absence of complexation. *The mass number corresponds to the monocation [**16**–H]$^+$

(Supplementary Fig. 45), the enhanced downfield shift of [12]CPP was exclusively attributed to the formation of ternary complex [12]CPP ⊃ (TQ•H$^+$/coronene) **17**, which was supported by ESI-TOF MS measurements (Fig. 5d, Supplementary Fig. 31). As such, TQ•TFA plays a pivotal role as a liaison to assemble [12]CPP and coronene, an inherently non-associative pair of molecules, via simultaneous π–π and CH–π interactions. The corresponding theoretical calculations suggest that the deformation energy of the apparently distorted [12]CPP ($\Delta E = 3.1$ kcal mol$^{-1}$) is compensated by a large negative association energy ($\Delta E = -81.7$ kcal mol$^{-1}$ after BSSE correction) (Supplementary Fig. 64).

**π–π interactions in aqueous phase.** By taking advantage of the water miscibility of TQ•TFA **10**, its π-stacking properties were further assessed in aqueous media by topoisomerase assays (Fig. 6b, Supplementary Methods). The potential applicability of **10** as a DNA intercalator was evaluated against the ability of topoisomerase I to relax supercoiled plasmid DNA. TQ•TFA **10** exhibited an inhibitory effect in a concentration-dependent fashion that is comparable to that of doxorubicin[64], a proven intercalator clinically used for cancer chemotherapy, which indicates that TQ•TFA **10** may be a potent intercalator for DNA. This study further revealed that the inhibitory properties of TQ•TFA **10** are distinct from those of structurally familiar DQ-

Im•TFA **9**, whereby the latter failed to inhibit the DNA relaxation under otherwise identical conditions, which could likely be attributed to a higher conformational flexibility that decreases its propensity towards intercalation.

## Discussion

We have designed and synthesised TriQuinoline (TQ), a 2D miniaturised molecular model for an atomic-sized defect surrounded by pyridinic-nitrogen sites in the structure of graphene. The strategic use of a cyclic DiQuinoline imine (DQ-Im) as the key intermediate was crucial for the high-yielding and reliable access to this unique flat molecule. DQ-Im was subjected to a Povarov reaction, i.e., a formal [4 + 2] cycloaddition reaction with enol ethers, to construct alkoxy tetrahydroquinoline, which resulted in unexpected sequential reactions under mild conditions that culminated in the successful synthesis of TQ. DFT calculations suggested that this unusual non-stop behaviour likely originates from the flat molecular topology, which facilitates the subsequent elimination and hydride transfer processes. TQ exhibits remarkably high proton affinity, whereby the exchange of the captured proton is kinetically hindered. TQ•H$^+$ is amenable to the formation of supramolecular complexes via plane-to-plane and edge-to-plane contact modes. The formation of binary complexes of plane-to-plane TQ•H+/coronene and edge-to-plane inclusion complexes [12]cycloparaphenylene(CPP) ⊃

**a**

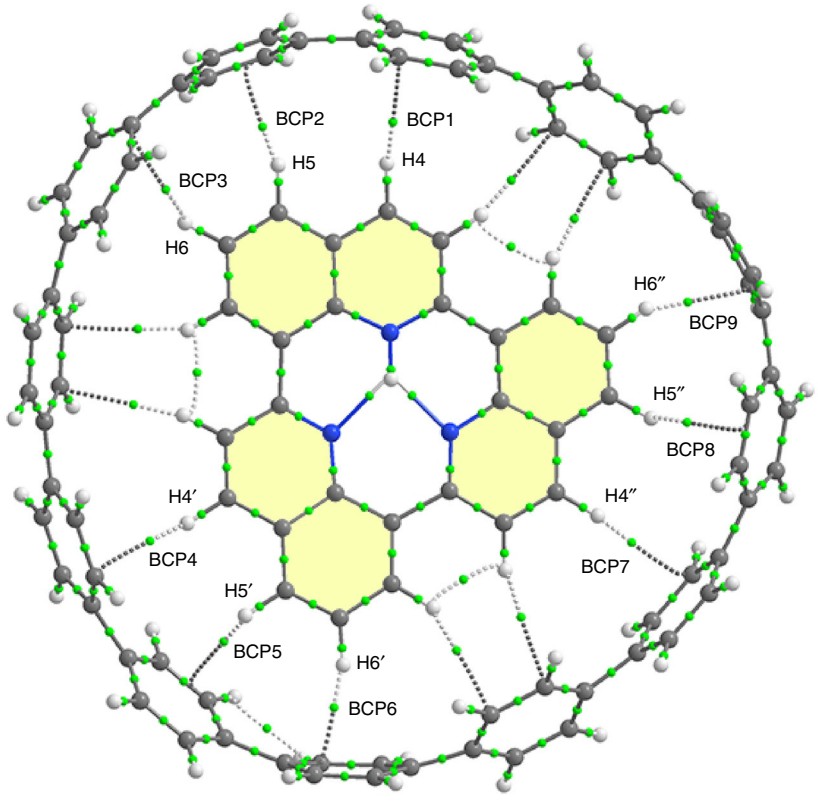

| BCP | H atom of TQ | $\rho$ (r$_c$) | $\nabla^2\rho$ (r$_c$) | $|\lambda_1|/\lambda_3$ |
|------|------|------|------|------|
| BCP1 | H4 | 0.007 | 0.021 | 0.19 |
| BCP2 | H5 | 0.008 | 0.030 | 0.20 |
| BCP3 | H6 | 0.010 | 0.027 | 0.17 |
| BCP4 | H4′ | 0.008 | 0.025 | 0.19 |
| BCP5 | H5′ | 0.007 | 0.024 | 0.17 |
| BCP6 | H6′ | 0.010 | 0.031 | 0.20 |
| BCP7 | H4″ | 0.008 | 0.023 | 0.19 |
| BCP8 | H5″ | 0.008 | 0.025 | 0.18 |
| BCP9 | H6″ | 0.009 | 0.031 | 0.19 |

**b**

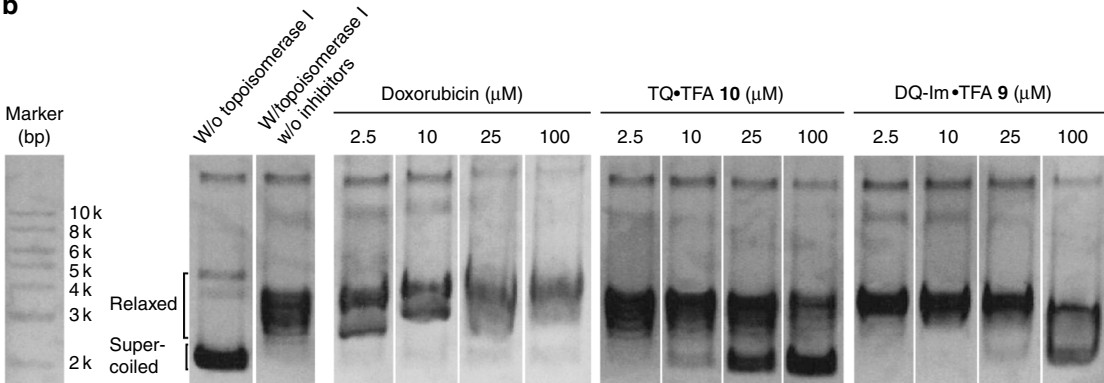

**Fig. 6 Non-covalent interactions of TQ. a** QTAIM analysis of the optimised structure of the inclusion complex [12]CPP ⊃ TQ•H$^+$ **15**. The [3, –1] bond critical points (BCPs) are shown as green circles on the bonds. The key parameters of the nine intermolecular BCPs that satisfy the criteria for CH–π interactions are shown: the electron density $\rho$(r$_c$), the Laplacian of the electron density ($\nabla^2\rho$(r$_c$)), and the ratio of the curvature of the density at the BCP ($|\lambda_1|/\lambda_3$). **b** Intercalation assays with topoisomerase I. Supercoiled DNA exhibits higher mobility (downward) in the gel. Doxorubicin and TQ•TFA **10** inhibit the relaxation of plasmid DNA induced by topoisomerase I in a concentration-dependent manner, while DQ-Im•TFA **9** exhibits a significantly weaker inhibitory effect. The uncropped gels are shown in Supplementary Fig. 68

$(TQ \cdot H^+)_{n(1-2)}$ was supported by NMR and ESI-TOF measurements and corroborated by computational calculations. The two-directional non-covalent interactions of $TQ \cdot H^+$ enabled the construction of the ternary complex [12]CPP ⊃ ($TQ \cdot H^+$/coronene), in which TQ serves as a liaison to assemble the inherently non-associative molecules [12]CPP and coronene. Moreover, the non-covalent interactions of water-miscible $TQ \cdot H^+$ are also operative in aqueous media, as confirmed by intercalation assays. Despite their considerable structural similarity, the remarkable features of TQ are absent in DQ-Im, further highlighting the significant impact of the rigid 2D architecture of TQ. These intriguing properties are revealed by the strategic bottom-up synthesis of TQ as a discrete small molecular mimic for a defective graphitic sheet featuring pyridinic nitrogens, allowing for an in-depth understanding of its inherent physicochemical properties. Building up polymeric entities including TQ units as well as potential analogues with distinct defect sizes and topologies will provide more opportunity to identify hitherto unknown functional materials, and as such is currently underway in our laboratory.

## Methods

**Materials**. Unless otherwise noted, materials were obtained from the commercial suppliers and used without further purification. THF, $CH_2Cl_2$, and $CH_3CN$ were purified by passing through a solvent purification system. Thin layer chromatography (TLC) was performed on glass TLC plates (0.25 mm) with silica gel 60 F254 and visualised by UV quenching and staining with $KMnO_4$. Flash column chromatography was performed using silica gel (neutral, spherical, 50–60 μm) or CombiFlash systems with a Redisep column. See Supplementary Methods for experimental details.

**Characterisations**. Infrared (IR) spectra were recorded on a HORIBA FT210 Fourier transform infrared spectrophotometer. NMR spectra were recorded on a Bruker AVANCE III HD400 or a JEOL ECZ-600R spectrometer. Chemical shifts (δ) are given in p.p.m. relative to residual solvent peaks. High-resolution mass spectra were measured on a ThermoFisher Scientific LTQ Orbitrap XL. ESI-TOF-MS spectra of supramolecular complexes were measured on a Bruker micrOTOF-II implemented with a CryoSpray source. Full characterisation data are supplied in Supplementary Methods with $^1H$, $^{13}C$, and $^{19}F$ NMR spectra (Supplementary Figs. 69–97).

**Computational study**. All quantum chemical calculations were performed using the Gaussian 16 programme[35]. DFT calculations employed an ultrafine integral grid (99 radial shells, 590 angular points). Structural optimisations were conducted at the level of theory specified in Supplementary Methods and the optimised geometries are supplied in Supplementary Tables 5–33. Frequency calculations confirmed the identity of geometry minima (no imaginary frequencies) and transitions states (one imaginary frequency). All transition-state structures were verified to connect the reactant and the product of interest by performing IRC calculations. Zero-point energies and thermal corrections were obtained at 298 K and are unscaled.

**Synthesis of 3**. A pressure-proof glass vial capable of being sealed with a Teflon cap was charged with 2,8-dichloroquinoline **1** (668 mg, 3.37 mol, 1.0 equiv), 2-(t-butoxycarbonylamino)phenylboronic acid **2** (800 mg, 3.37 mol, 1.0 equiv), $K_2CO_3$ (3.50 g, 25.3 mmol, 7.5 equiv), and $Pd(PPh_3)_4$ (204 mg, 0.176 mmol, 5.0 mol%) and was evacuated and back-filled with argon. To the mixture, 1,4-dioxane (12.7 mL) and $H_2O$ (5.9 mL) were added at 0 °C, and the vial was evacuated, back-filled with argon for several times, and sealed. The reaction mixture was stirred at 85 °C for 15 h. The resulting mixture was diluted with EtOAc/$H_2O$ and extracted three times with EtOAc. The combined organic layer was dried over $Na_2SO_4$, filtered, and concentrated. The residue was purified by flash column chromatography (n-hexane/EtOAc, 100/0–5.7/1) to give **3** (589 mg, 49% yield).

**Synthesis of 6**. To an oven dried pressure-proof glass vial capable of being sealed with a Teflon cap were added XPhos Pd G2 (9.97 mg, 0.0127 mmol, 1.0 mol%), XPhos (12.1 mg, 0.0253 mmol, 2.0 mol%), tetrahydroxydiboron (170 mg, 1.90 mmol, 1.5 equiv), KOAc (373 mg, 3.80 mmol, 3.0 equiv), and **3** (450 mg, 1.27 mmol, 1.0 equiv). The vial was sealed, evacuated, and filled with argon (four times). EtOH (12.7 mL) was added at 0 °C via syringe followed by the addition of ethylene glycol (212 μL 3.80 mmol, 3.0 equiv). The mixture was stirred at 80 °C for 1 h. After cooling the mixture to room temperature and removing the seal, $K_2CO_3$ (263 mg, 1.90 mmol, 1.50 equiv), $PPh_3$ (9.97 mg, 0.0380 mmol, 3.0 mol%), and $Pd(PPh_3)_4$ (73.2 mg, 0.0633 mmol, 5.0 mol%) were added. After adding $H_2O$ (3.68 mL) via a syringe, the

mixture was left stand for 5 min. After subsequent addition of 2-chloro-8-methylquinoline **5** (248 mg, 1.39 mmol, 1.1 equiv), the vial was sealed, evacuated, and back-filled with argon (four times). The resulting mixture was stirred at 83 °C for 24 h. The reaction mixture was diluted with EtOAc/$H_2O$ and extracted three times with EtOAc. The organic layer was dried over $Na_2SO_4$, filtered, and concentrated. The residue was purified by flash column chromatography (n-hexane/EtOAc, 100/0–3/1) to give **6** (385 mg, 66% yield).

**Synthesis of TQ•TFA 10**. To a pressure-proof glass vial capable of being sealed with a Teflon cap were added **6** (40.5 mg, 0.0877 mmol, 1.0 equiv), N-bromo-succinimide (NBS, 38.7 mg, 0.218 mmol, 2.5 equiv), azobisisobutyronitrile (AIBN, 2.55 mg, 0.0155 mmol, 0.18 equiv), and carbon tetrachloride (2.0 mL). The vial was flushed with argon and sealed. The reaction mixture was stirred at 77 °C for 5 h. The resulting mixture was dissolved in $CH_2Cl_2$ and subsequently washed with 1 N NaOH (aq.), $H_2O$, and brine. The organic layer was dried over $Na_2SO_4$, filtered, and concentrated to afford crude **7** in high purity. To a solution of crude **7** in degassed dioxane (9.0 mL) in a round-bottomed flask, degassed 10% $Na_2CO_3$ aq. (10.0 mL) was added. The flask was then equipped with a reflux condenser fitted with a three-way stopcock opened to an argon-filled balloon. The reaction mixture was stirred at 80 °C for 16 h under argon. The resulting mixture was diluted with EtOAc/$H_2O$ and extracted three times with EtOAc. The combined organic layer was dried over $Na_2SO_4$, filtered, and concentrated to afford crude **8** in high purity, which was used in the next reaction without further purification. To a round-bottomed flask containing crude **8** was added 1,2-dichloroethane (7.42 mL) and trifluoroacetic acid (TFA, 67.2 μL, 0.877 mmol, 10 equiv). The flask was then equipped with a reflux condenser fitted with a three-way stopcock opened to an argon-filled balloon. The reaction mixture was stirred at 70 °C for 18 h under argon. The resulting mixture was concentrated and dissolved in water (16.4 mL). The aqueous solution was lyophilised to afford crude DQ-Im•TFA **9** in high purity. To a round-bottomed flask containing crude DQ-Im•TFA **9** was added $CH_3CN$ (8.35 mL) and butyl vinyl ether (BVE, 114 μL, 0.877 mmol, 10 equiv). The reaction mixture was stirred at 45 °C for 25 h. The resulting mixture was concentrated and dissolved in water, and washed with $CH_2Cl_2$ (eight times). The aqueous layer was lyophilised to afford **10** in pure form (19.9 mg, 46% yield over four steps from **6**).

## Data availability

The X-ray crystallographic coordinates for structures reported in this study have been deposited at the Cambridge Crystallographic Data Centre (CCDC), under deposition numbers CCDC-1909996 (compound **6**) and CCDC-1909994 (compound **S4**). These data can be obtained free of charge from the CCDC via www.ccdc.cam.ac.uk/data_request/cif. All other data support findings of this study are available from the corresponding author upon reasonable request.

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

## Acknowledgements

This work was financially supported by a Grant-in-Aid for Scientific Research B (KAKENHI 17H03025) and Precisely Designed Catalysts by Customised Scaffolding (KAKENHI JP16H01043) from the JSPS. Dr. Naoki Takizawa is gratefully acknowledged for carrying out the topoisomerase I assays. The authors thank Dr. Tomoyuki Kimura for the X-ray crystallographic analysis of compounds **6** and **S4**, as well as Dr. Ryuichi Sawa, Ms. Yumiko Kubota, and Dr. Kiyoko Iijima for technical assistance with the NMR analyses. Part of the computation resources in this work was provided by the Research Centre for Computational Science, Okazaki, Japan.

## Author contributions

N.K. conceived the project and designed TriQuinoline. N.K. and M.S supervised the study. N.K. and S.A. designed the experiments. S.A. performed the experiments. N.K. wrote the manuscript with revision provided by S.A. and M.S. All authors discussed the results and commented on the manuscript.

## Additional information

**Competing interests:** The authors declare no competing interests.

