## [Peer Review File · Nature Communications]

Reviewers' Comments:

Reviewer #1:

Remarks to the Author:

This manuscript reports the title heterocyclic compound as a proton capture as a novel defect-containing graphene-related compound. Although only one compound is treated here, it has interesting properties and show phenomena in various fields. The main sections consist of synthesis, reaction mechanism, properties involving proton affinity, performance as guest molecules, and DNA intercalator.

This manuscript is potentially publishable in Nature Commun. after minor revisions and if the authors can reply to the following comments reasonably.

- a. As for the synthesis and mechanism, the authors adopted a Povarov approach and the reaction nature was explored by some experimental and theoretical methods. Although the data were well collected and elucidate, the proposed reaction course seems to be rather usual and a combination of routine reactions.
- b. The basicity should be a key performance of this heterocyclic compound. Although the author failed to deprotonate it by treatment with strong bases, the use of super bases (Chem. Rev. 1993, 93, 2317; Book "Superbases for Organic Synthesis") may affect the deprotonation. A competitive experiment with other strong bases to monitor proton exchange process possible determines an estimated value of pKa. The properties of TQ should be compared with aminopyridine macrocycles reported by Kanbara et al (ref. 22, and others such as EJOC 2006, 3314) because of the close structural similarity.
- c. Recently, CPP derivatives are known to include electron efficient aromatic compounds such as pyridinium salts (for example JOC 2017, 82, 9885). Therefore, the inclusion of TQ by [12]CPP is not a surprising phenomenon. Its association constant is not so high. It is notable that TQ forms complex in 1:2 ratio (complex 16): are there any interactions between TQ molecules?

Other specific comments:

- d. The properties of TQ salts should be affected by the kind of counter anion and solvents.
- e. A weak point is that no X-ray data of TQ itself and its complexes are given. The calculated structures in Fig. 5 are reasonable, but other energy-minimum structures are possible.
- f. In order to determine the complexation ratio, Job's plots are often adopted.

Reviewer #2:

Remarks to the Author:

The authors describe the challenging synthesis of an exciting new macrocycle, along with investigations on the reaction mechanism for the synthesis, the proton-sponge-type basicity of the new compound and the formation of supramolecular unusual complexes. The paper is very well-written and the research has been done very meticulously. This becomes already evident in Fig. 1c where the authors were not satisfied with an unreliable low-yielding synthesis, but developed a much more robust and original route instead. Irrespective of the claim that the new compound is a mimick of heteroatom-doped graphitic materials, there are three aspects that make this study worthy of publication in Nature Communications: 1) The authors demonstrate very convincingly (experiment & theory) that the IED-hetero-Diels-Alder reaction bypasses the usual (THQ, DHQ) intermediates and that an unusual hydride transfer in a pi-stacked supramolecular complex occurs. In my view, this shows very clearly how unique TQ is as a synthetic target and how much one can learn about different facets of "aromaticity" from such compounds. 2) TQ appears to be extremely (!) Bronsted basic and indeed the salt TQ*H⁺ cannot be deprotonated easily or without decomposition. This fact sheds some doubt on the validity of the paper's title (the authors did not isolate or investigate TQ; moreover, the results indicate that TQ is most likely not a stable molecule), but this does not diminish the

importance of the paper, because TQ*H⁺ is (partially) water soluble and a more interesting platform for supramolecular chemistry. 3) very elegant studies have been performed on forming non-covalent complexes with flat coronene, nanohoop [12]CPP and DNA (while only preliminary, suggesting potential medical uses of TQ*H⁺).

Suggestions for improvement:

- Consider changing the title (see above).
- Fig. 1c: consider adding a yield (range) to the reaction arrow on the right hand side
- 5.7% yield: consider changing this to 6%, unless the accuracy is really that high (balances etc.). In any case though, if the reaction is not reproducible, "5.7%" sends the wrong message to readers.
- Double-check if the term "unsaturation" is correctly used (I only know it from the concept of "degree of unsaturation", not as a reaction type)
- Page 8 – attempts to deprotonate TQ*H⁺: the authors should mention, which bases were used in the "numerous attempts" to deprotonate TQ*H⁺ and whether the decomposition products correspond to smaller or larger fragments (simple mass spec). In my view, the authors should also more clearly state, if their conclusion of these studies is that TQ is not a stable molecule (under standard conditions). From the list of bases studied, one could most likely make some estimates on the minimum pKa of TQ*H⁺ (only strong enough bases should lead to deprotonation and ensuing deprotonation).
- Figure 4b: "TQ-Im*TFA 10" (pink font) is most likely incorrect
- Why was no association constant determined for the binary complex with coronene (K_a too low for NMR determination)? The reason should at least be mentioned. For the ternary complex, I presume it would be difficult to perform an adequate titration?
- The supramolecular results should perhaps be discussed in comparison with the most closely related example in the literature (ref. 45). A recent review in this field could be of help:
<https://onlinelibrary.wiley.com/doi/10.1002/anie.201906069>
- Conclusion section: If the authors regard TQ as a mimick for graphitic materials, they should discuss any implications of their work in this context here

Response to reviewers

–Reviewer #1–

This manuscript reports the title heterocyclic compound as a proton capture as a novel defect-containing graphene-related compound. Although only one compound is treated here, it has interesting properties and show phenomena in various fields. The main sections consist of synthesis, reaction mechanism, properties involving proton affinity, performance as guest molecules, and DNA intercalator.

This manuscript is potentially publishable in Nature Commun. after minor revisions and if the authors can reply to the following comments reasonably.

Reviewer #1 a. as for the synthesis and mechanism, the authors adopted a povarov approach and the reaction nature was explored by some experimental and theoretical methods. although the data were well collected and elucidate, the proposed reaction course seems to be rather usual and a combination of routine reactions.

Answer We admit that the quinoline formation via cycloaddition looks straightforward. However, as reviewer #2 noted, the expeditious formation of the quinoline ring without the identification of the tetrahydroquinoline and dihydroquinoline intermediates is unusual, and it is closely related to the structural feature of TriQuinoline as described in the manuscript. We believe that this unusual reactivity deserves broad publicity.

Reviewer #1 b. The basicity should be a key performance of this heterocyclic compound. Although the author failed to deprotonate it by treatment with strong bases, the use of super bases (Chem. Rev. 1993, 93, 2317; Book “Superbases for Organic Synthesis”) may affect the deprotonation. A competitive experiment with other strong bases to monitor proton exchange process possible determines an estimated value of pKa. The properties of TQ should be compared with aminopyridine macrocycles reported by Kanbara et al (ref. 22, and others such as EJOc 2006, 3314) because of the close structural similarity.

Answer This is indeed the critical point to illuminate the difference of TriQuinoline and the compounds of similar structure. Prof. Kanbara’s compounds, trimers of pyridine connected by three nitrogen atoms, can exist as free amines, be more flexible, and behave as conventional bases. Our TriQuinoline is a much more rigid chemical entity and this feature endows TriQuinoline with completely different chemical properties. Prof. Kanbara’s compounds work as base catalysts, meaning that their compounds can capture and release protons quickly, whereas kinetic reluctance of the proton in TriQuinoline is simply unusual and originates from its rigid structure.

The key difference is: our TriQuinoline captures proton with unusual thermodynamic stability (even outperforming proton sponge) and Kanbara’s compounds behaves as normal strong bases. It is clear from the fact that Kanbara’s compounds can be isolated in the organic phase and behave as a regular organic molecular entities. In sharp contrast, our TriQuinoline has no chance to be isolated as a free base and virtually insoluble in most organic solvents. We believe that these facts sufficiently represent the explicit difference of these molecules, which is originated from the simple but unique structure of TriQuinoline. TriQuinoline displayed further exclusive behaviour e.g. supramolecular complexation, and intercalation.

Reviewer #1 c. Recently, CPP derivatives are known to include electron efficient aromatic compounds such as pyridinium salts (for example JOC 2017, 82, 9885). Therefore, the inclusion of TQ by [12]CPP is not a surprising phenomenon. Its association constant is not so high. It is notable that TQ forms complex in 1:2 ratio (complex 16): are there any interactions between TQ molecules?

Answer As this reviewer kindly pointed out, a [8]CPP derivative has been reported to form an inclusion complex with a pyridinium cation via CH- π interactions. We have included the suggested article in the references (ref 41). However, it is operative only when one of the Ph ring of [8]CPP is replaced with an electron-rich 1,4-dimethoxybenzene to engage complexation. The failure of DQ-Im•H⁺ to form an inclusion complex with [12]CPP underscores that the importance of the delicate balance of electronic nature of both host and guest, as well as the number of peripheral protons of the guest. In DFT calculations, the plane-to-plane distance of two TQ molecules is 3.3 Å, which implies that the attractive π - π interactions are operative, which was also supported by NCI plot.

Reviewer #1 d. The properties of TQ salts should be affected by the kind of counter anion and solvents.

Answer As the reviewer pointed out, we had changed the counter anion from trifluoroacetate to triflate and dodecane carboxylate, with the former having slightly better crystalline nature and the latter having better solubility in organic solvents. However, despite extensive efforts, an analytically pure sample was not obtained even after HPLC purification, and thus we presented the results concerning the TFA salt of TQ. This salt was mostly insoluble to most organic solvents, and sparingly soluble in acetonitrile and DMSO. Therefore, our only option was to use DMSO to study supramolecular complexation with nonpolar hydrocarbons such as coronene and [12]CPP.

Reviewer #1 e. A weak point is that no X-ray data of TQ itself and its complexes are given. The calculated structures in Fig. 5 are reasonable, but other energy-minimum structures are possible.

Answer We have no objection on this comment. Indeed, we have made considerable efforts to crystallize TQ (salts) and its complexes, but all of these attempts were unfortunately fruitless. We believe that the rapidly emerging microED method will be the standard for the solid state structural determination in the near future, in particular for these kinds of poorly crystalline compounds. To compensate for the lack of the crystal structure, we believe that we have done the best possible empirical analyses to characterize TQ and its complexes. We would appreciate it if the reviewer and the editor could empathize with our situation. As for the calculated complex structures, after extensive geometry optimizations from distinct initial geometries, all attempts converged on the structures presented for TQ/coronene 14 and TQ/[12]CPP complexes 15 and 16, likely due to the rigid nature of these molecules (For 15, even starting from the perpendicular topology of TQ and [12]CPP, the structure converged to the spoke and rim topology). For the ternary complex TQ/coronene/[12]CPP 17, the presented structure was the only converged structure and most of the attempts failed to locate the stationary point.

Reviewer #1 f. In order to determine the complexation ratio, Job's plots are often adopted.

Answer In recent years, Job's method is under scrutiny and is being questioned as a reliable analytical method (For example, P. Thordarson, *Chem. Commun.* 2016, 52, 12792 "The death of the Job plot, transparency, open science and online tools, uncertainty estimation methods and other developments in supramolecular chemistry data analysis" and J. Jurczak, *J. Org. Chem.* 2016, 81, 1746 "Recognizing the limited applicability of Job plots in studying host-guest interactions in supramolecular chemistry"). For this reason we took advantage of the non-linear regression analysis reported by Thordarson to more precisely and empirically determine the complexation ratio. Indeed, this analysis is well exploited to determine complexation ratios e.g. *J. Am. Chem. Soc.* 2017, 139, 18496, and *Chem. Sci.* 2018, 9, 3477. We believe that this analysis is the best possible method to analyse supramolecular complexation for this system.

–Reviewer #2–

The authors describe the challenging synthesis of an exciting new macrocycle, along with investigations on the reaction mechanism for the synthesis, the proton-sponge-type basicity of the new compound and the formation of supramolecular unusual complexes. The paper is very well-written and the research has been done very meticulously. This becomes already evident in Fig. 1c where the authors were not satisfied with an unreliable low-yielding synthesis, but developed a much more robust and original route instead. Irrespective of the claim that the new compound is a mimick of heteroatom-doped graphitic materials, there are three aspects that make this study worthy of publication in Nature Communications: 1) The authors demonstrate very convincingly (experiment & theory) that the IED-hetero-Diels-Alder reaction bypasses the usual (THQ, DHQ) intermediates and that an unusual hydride transfer in a pi-stacked supramolecular complex occurs. In my view, this shows very clearly how unique TQ is as a synthetic target and how much one can learn about different facets of "aromaticity" from such compounds. 2) TQ appears to be extremely (!) Bronsted basic and indeed the salt $TQ \cdot H^+$ cannot be deprotonated easily or without decomposition. This fact sheds some doubt on the validity of the paper's title (the authors did not isolate or investigate TQ; moreover, the results indicate that TQ is most likely not a stable molecule), but this does not diminish the importance of the paper, because $TQ \cdot H^+$ is (partially) water soluble and a more interesting platform for supramolecular chemistry. 3) very elegant studies have been performed on forming non-covalent complexes with flat coronene, nanohoop [12]CPP and DNA (while only preliminary, suggesting potential medical uses of $TQ \cdot H^+$).
Suggestions for improvement:

Reviewer #2 Consider changing the title (see above).

Answer We admit that a free base of TriQuinoline as a molecular entity is possibly not stable enough, and $TQ \cdot H^+$ indeed predominately appears in this manuscript. However, the underlying chemistry described in this manuscript is necessarily linked to the unique TriQuinoline architecture. We believe that 'TQ: TriQuinoline' is the best title for the present work (although we seriously considered and deliberated this point before the initial submission). We believe that this short title would be more attractive to potential readers, and would appreciate if the reviewer and editor could agree with the title.

- Reviewer #2 Fig. 1c: consider adding a yield (range) to the reaction arrow on the right hand side.
- Answer We thank this reviewer for this idea to highlight the difference of synthetic efficiency. We fulfilled this suggestion in the revised Figure.
- Reviewer #2 5.7% yield: consider changing this to 6%, unless the accuracy is really that high (balances etc.). In any case though, if the reaction is not reproducible, "5.7%" sends the wrong message to readers.
- Answer We agree with this opinion. We have changed '5.7%' to '6%'.
- Reviewer #2 Double-check if the term "unsaturation" is correctly used (I only know it from the concept of "degree of unsaturation", not as a reaction type).
- Answer This comment is quite reasonable in strict definition. We have replaced the word 'unsaturation' with 'aromatisation' in Line 46, Page 3.
- Reviewer #2 Page 8 – attempts to deprotonate TQ*H+: the authors should mention, which bases were used in the "numerous attempts" to deprotonate TQ*H+ and whether the decomposition products correspond to smaller or larger fragments (simple mass spec). In my view, the authors should also more clearly state, if their conclusion of these studies is that TQ is not a stable molecule (under standard conditions). From the list of bases studied, one could most likely make some estimates on the minimum pKa of TQ*H+ (only strong enough bases should lead to deprotonation and ensuing deprotonation).
- Answer We agree that the information of the bases used helps readers to know the proton affinity and base compatibility of TQ•H⁺ more precisely. We attempted to extract the TQ free amine by partition with sat. NaHCO₃ aq and 1N NaOH aq., but nothing was extracted to the organic layer and TQ decomposed based on MS (Mw similar or smaller). When TQ•TFA in CD₃CN was treated with organic bases e.g. Et₃N and tert-butylimino-tri(pyrrolidino)phosphorene, however consequently giving a complicated mixture. We have included the following additional information in Line 141. (e.g. aqueous phase: NaOH, in acetonitrile: from weak triethylamine to strong *tert*-butylimino-tri(pyrrolidino)phosphorene)
- Reviewer #2 Figure 4b: "TQ-Im*TFA 10" (pink font) is most likely incorrect
- Answer We thank this reviewer for pointing out the typo. We have corrected accordingly (from "TQ-Im•TFA 10" to "TQ•TFA 10").
- Reviewer #2 Why was no association constant determined for the binary complex with coronene (Ka too low for NMR determination?)? The reason should at least be mentioned. For the ternary complex, I presume it would be difficult to perform an adequate titration?

Answer In fact we had shown K_a for TQ/coronene complex 14 in Figure 5 in the original submission. As for the ternary complex, as pointed out, we believe that it is not technically feasible.

Reviewer #2 The supramolecular results should perhaps be discussed in comparison with the most closely related example in the literature (ref. 45). A recent review in this field could be of help: <https://onlinelibrary.wiley.com/doi/10.1002/anie.201906069>.

Answer We thank this reviewer for suggesting to consult this recently released comprehensive review. We have included this review article as ref 47 and revised the sentence to note the difference of TQ/[12]CPP and the example of ref 45 (former numbering).

before

“The uniform upfield shift of the proton resonance of TQ•H⁺ is thereby consistent with the previous literature on the structurally-related CH– π -driven inclusion complexes of naphthalene-walls and corannulene.⁴⁵”

After

“The uniform upfield shift of the proton resonance of TQ•H⁺ is thereby consistent with the previous literature on the structurally-related CH– π -driven inclusion complex. Isobe et al. reported an inclusion complex featuring a bowl-in-tube topology comprising corannulene (bowl) and cyclic chrysene tetramer (tube), where 10 CH– π hydrogen bonds are responsible for the formation of the inclusion complex.^{46,47}”

Reviewer #2 Conclusion section: If the authors regard TQ as a mimick for graphitic materials, they should discuss any implications of their work in this context here.

Answer We thank this reviewer for the productive suggestion. We have added the following sentence to the conclusion section.

“These intriguing properties are revealed by the strategic bottom-up synthesis of TQ as a discrete small molecular mimic for a defective graphitic sheet featuring pyridinic-nitrogens, allowing for an in-depth understanding of its inherent physicochemical properties. Building up polymeric entities including TQ units as well as potential analogues with distinct defect sizes and topologies will provide more opportunity to identify hitherto unknown functional materials, and as such is currently underway in our laboratory.”